# RoSE: Enhancing SE(3)-based Protein Backbone Generation via Robust Score Estimation

## Abstract

This work presents improvements to Riemannian diffusion models for protein structure generation by developing robust heat kernel computation methods on SE(3) space. While existing approaches suffer from approximation errors in score-based diffusion, our method enables stable and accurate denoising score matching on the high-dimensional $SE(3)^N$ manifold through theoretically-grounded numerical techniques. The proposed framework achieves competitive performance in protein generation benchmarks, demonstrating superior scores and successfully generating diverse, physically-plausible protein structures. Notably, our model solves 23 out of 24 motif scaffolding problems and designs refoldable nanobodies, significantly advancing the capability to generate functional protein geometries while maintaining mathematical consistency with the underlying manifold structure.

## 1 Introduction

While diffusion models have achieved remarkable success in modeling data distributions within Euclidean space (Ho et al., 2020; Song et al., 2021; Saharia et al., 2022), their direct application to scientific domains like high-energy physics (Brehmer & Cranmer, 2020), geological science (Karpatne et al., 2019), and computational biology (Wu et al., 2022b) often yields suboptimal performance (Bortoli et al., 2022). This limitation stems from a fundamental geometric mismatch: such scientific data is normally best represented on complex manifolds, since directly applying Euclidean diffusion models does not properly incorporate the data prior, and training such model often suffers from singularities on these complex manifolds (Lou et al., 2023).

In particular, *de novo* protein design – the task of generating novel proteins satisfying specified structural or functional properties – faces the same challenge due to the inherent complexity of the data distribution (Levinthal, 1969), which resides on a highly intricate high-dimensional manifold: a protein backbone consists of $N$ residues, each with four heavy atoms rigidly connected via covalent bonds. Since each residue can be described as an element of the Lie group $SE(3)$ (Jumper et al., 2021; Yim et al., 2023c), the structure space of protein backbones forms a high-dimensional Riemannian manifold, formally modeled as $SE(3)^N$. To model such data faithfully, generative methods like diffusion processes must operate directly on this manifold rather than in Euclidean space. Thus, a significant challenge lies in formulating diffusion processes that rigorously adhere to the $SE(3)^N$ manifold's geometric priors.

Among various attempts to model protein structures on their manifolds, the $SE(3)$ score-based diffusion model (Huang et al., 2022; Yim et al., 2023c; Bortoli et al., 2022) has emerged as a promising solution. Several protein generation methods(Watson et al., 2022; Trippe et al., 2023) have adopted this framework, learning to reverse $SE(3)^N$ diffusion process – in particular, the $SO(3)$ and $\mathbb{R}^3$ heat equation for each residue respectively. Although such methods have generated experimentally verified and novel protein binders (Watson et al., 2022), their training process encounters instabilities arising from score approximation errors on $SE(3)^N$. Thus, such works resort to noise schedule truncation, which restricts training noise level $\sigma \geq \sigma_{truncated}$. This results in undesignable samples including chain breaks or steric clashes(See Fig. 2). Recent methods like RFDiffusion (Watson et al., 2022) and FrameDiff (Yim et al., 2023c) either introduce heuristic loss or rely on pretraining on protein structure prediction to alleviate physical violations in generated samples. However, such practical improvements not only introduce additional hyperparameter tuning but also fundamentally introduce biases to the generative distributions (Liu et al., 2024).

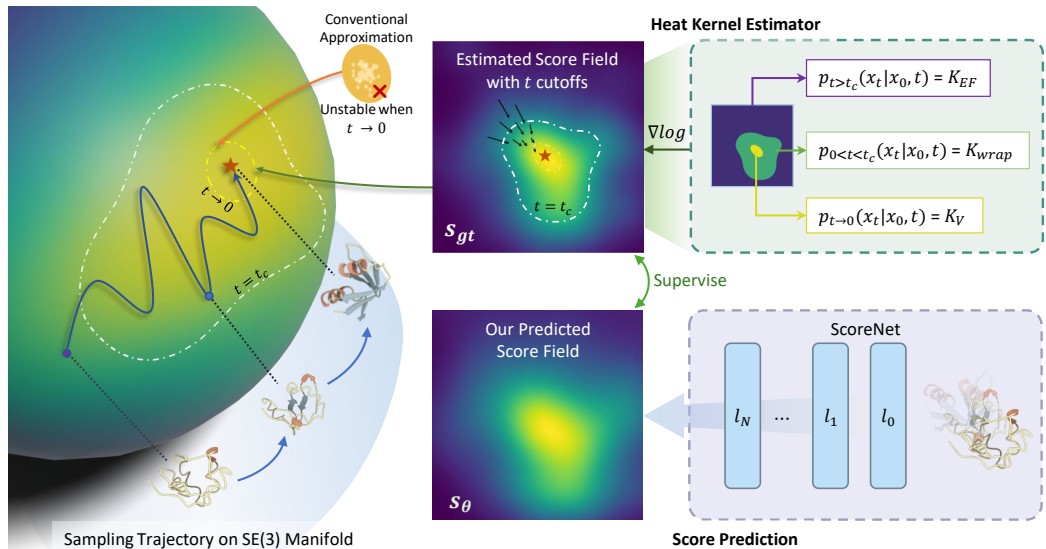

Figure 1: **High-quality protein backbone generation via multi-stage score estimation.** Conventional approximations break down when noise levels are low, making accurate score-function simulation challenging. Our method, RoSE, leverages a multi-stage heat kernel estimator to construct a precise score field, enabling the network to more faithfully learn the true data distribution. As a result, RoSE generates protein backbones of substantially higher quality.

In this work, we introduce a strategy to stabilize Riemannian Diffusion Models (RDMs) by re-examining the computational foundations of denoising score matching. Central to this framework is the heat kernel (Grigor'yan, 1999), which governs the diffusion process and enables

Table 1: Overview of various $SE(3)$-based protein backbone generation models applying strategies to mitigate score approximation errors.

| Method | Framework | Noise Schedule | Additional Loss |
|---|---|---|---|
| RFDiffusion (Watson et al., 2022) | RDM | truncated | required |
| FrameDiff (Yim et al., 2023c) | RDM | truncated | required |
| FOLDFLOW (Bose et al., 2024) | RFM | – | optional |
| FrameFlow (Yim et al., 2023a) | RFM | – | optional |
| Ours | RDM | full | optional |

gradient estimation for training. By exploiting the unique algebraic and leverage symmetry of $SO(3)$ space, we derive efficient numerical methods for both sampling from the heat kernel and computing the gradient of its logarithm, which are critical for stable denoising score matching. This strategy enables robust training of diffusion models on a complex $SE(3)^N$ manifold, a canonical space for protein structure representation.

Our contributions are threefold. First, we propose RoSE, a **Ro**bust **SE**(3) diffusion model that leverages Varadhan's asymptotic formula for the heat kernel tailored for protein design task. By combining a geodesic distance-based approximation with a wrapped summation over periodic copies, our method preserves numerical stability when the diffusion timestamp is small. Second, our architectural improvements substantially enhance model performance. Furthermore, the carefully curated dataset provides additional gains. Together, these advances enable the generation of biologically designable protein monomers with greater structural diversity and novel folding patterns. Third, we extend these capabilities to address a critical challenge in computational biology by solving motif scaffolding problems: our approach successfully scaffolds 23 out of 24 structural motifs in standard benchmarks and produces refoldable VHH nanobody designs for 4 out of 25 difficult cases, matching the current state-of-the-art RFDiffusion's success rate of 5 out of 25, thereby advancing the field of conditional protein design.

## 2 BACKGROUND AND PRELIMINARIES

### 2.1 PROTEIN BACKBONE PARAMETERIZATION

Diffusion models have been applied to various protein representations, including torsion angles (Wu et al., 2022a), $C_\alpha$-only coordinates (Geffner et al., 2025), and $SE(3)$ backbone frame representations (Watson et al., 2022). Among these, the frame representation achieves remarkable results

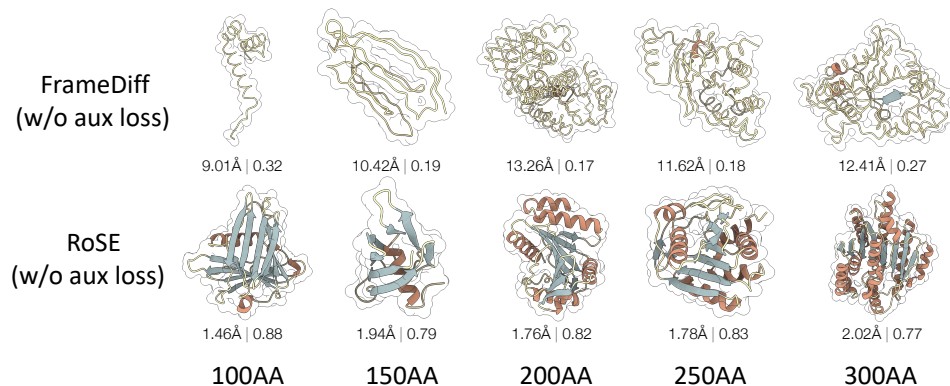

FrameDiff
(w/o aux loss)

9.01Å | 0.32    10.42Å | 0.19    13.26Å | 0.17    11.62Å | 0.18    12.41Å | 0.27

RoSE
(w/o aux loss)

1.46Å | 0.88    1.94Å | 0.79    1.76Å | 0.82    1.78Å | 0.83    2.02Å | 0.77

100AA        150AA        200AA        250AA        300AA

Figure 2: **Sampled backbones** from $\mathcal{IG}_{SO(3)}$ based diffusion model like FrameDiff and RoSE solely trained by score matching loss. First row shows the sampled backbones from $\mathcal{IG}_{SO(3)}$ based diffusion model, and the second row shows the sampled backbones from RoSE. When trained solely with the score-matching loss, RoSE produces backbones with fewer chain breaks and steric clashes than those generated by the $\mathcal{IG}_{SO(3)}$ based diffusion model. Below each sampled structure, we report scRMSD($\downarrow$) and scTM($\uparrow$).

in protein design tasks (Watson et al., 2022). Our protein backbone representation follows this frame-based approach. The 3D backbone structure of a protein with $N$ residues is represented by rigid transformations in SE(3), where each residue $i$ is associated with a transformation $T_i = (R_i, \mathbf{t}_i)$ consisting of a rotation matrix $R_i \in$ SO(3) and a translation vector $\mathbf{t}_i \in \mathbb{R}^3$. Applying $T_i$ to canonical coordinates $\{\mathbf{N}^\circ, \mathbf{C}_\alpha^\circ, \mathbf{C}^\circ\}$ (with $\mathbf{C}_\alpha^\circ = \mathbf{0}$) via

$$T_i \circ \mathbf{p} = R_i \mathbf{p} + \mathbf{t}_i \quad \forall \mathbf{p} \in \{\mathbf{N}^\circ, \mathbf{C}_\alpha^\circ, \mathbf{C}^\circ\} \tag{1}$$

generates observed atomic positions. The $\mathbf{O}$ atom positions are determined following AlphaFold2 (Lin et al., 2022) using an additional torsion angle per residue.

SE(3) is a special Euclidean space containing rotation and translation, which can be formally written as SE(3) $\cong$ SO(3) $\ltimes \mathbb{R}^3$, where $\ltimes$ is the semi-direct product operator. It means the rotation and translation components of SE(3) are independent of each other. Therefore, one can implement SE(3) diffusion models by learning the reverse diffuse process of SO(3) and $\mathbb{R}^3$ respectively under a chosen diffusion framework like Variance-Preserving score-based diffusion (VP-SDM) (Song et al., 2021). In particular, the forwarding stochastic process on SE(3)$^N$ is given by (Yim et al., 2023c):

$$\mathrm{d}\mathbf{X}_{\mathrm{SE}(3)^N}^{(t)} = \left[0, -\frac{1}{2}\mathbf{X}_{\mathbb{R}^{3N}}^{(t)}\right]\mathrm{d}t + \left[\mathrm{d}\mathbf{B}_{\mathrm{SO}(3)^N}^{(t)}, \ \mathrm{d}\mathbf{B}_{\mathbb{R}^{3N}}^{(t)}\right] \tag{2}$$

where $\mathrm{d}\mathbf{B}_\mathcal{M}^t$ is the natural analog of Brownian motion for the compact, differentiable manifold $\mathcal{M}$.

This stochastic differential equation (SDE) (Song et al., 2021) has a corresponding decomposed reversed SDE,

$$\mathrm{d}\overleftarrow{\mathbf{X}}_{\mathrm{SO}(3)^N}^{(t)} = \nabla_\mathbf{X} \log p_t\left(\overleftarrow{\mathbf{X}}_{\mathrm{SO}(3)^N}^{(t)}\right)\mathrm{d}t + \mathrm{d}\mathbf{B}_{\mathrm{SO}(3)^N}^{(t)}, \tag{3}$$

$$\mathrm{d}\overleftarrow{\mathbf{X}}_{\mathbb{R}^{3N}}^{(t)} = \left\{\frac{1}{2}\overleftarrow{\mathbf{X}}_{\mathbb{R}^{3N}}^{(t)} + \nabla_\mathbf{X} \log p_t\left(\overleftarrow{\mathbf{X}}_{\mathrm{SO}(3)^N}^{(t)}\right)\right\}\mathrm{d}t + \mathrm{d}\mathbf{B}_{\mathbb{R}^{3N}}^{(t)}. \tag{4}$$

Since the translation component lies in $\mathbb{R}^{3N}$ space, $\nabla_\mathbf{X} \log p_t(\overleftarrow{\mathbf{X}}_{\mathbb{R}^{3N}}^{(t)})$ can be thereby easy to derived (see Appendix. A) since the transition kernel (known as heat kernel), $\mathrm{d}\mathbf{B}_{\mathbb{R}^{3N}}^{(t)}$, corresponds to Gaussian. However, the complexity of Riemannian SO(3) manifold renders deriving $\nabla_\mathbf{X} \log p_t(\overleftarrow{\mathbf{X}}_{\mathrm{SO}(3)^N}^{(t)})$ not easy as the SO(3) heat kernel, $p_{\mathrm{SO}(3)}(x_t|x_0, t)$, has no closed form. To work with it, the current

SE(3)-based protein diffusion models have to rely on explicit expression of eigenfunctions $f_i$ on SO(3), which satisfy $\Delta f_i = -\lambda_i f_i$, to approximate heat kernel function with an infinite sum:

$$p_{SO(3)}(x_t|x_0, t) = \sum_i e^{-\lambda_i t} f_i(x_t, x_0) = \frac{1}{8\pi^2} \sum_{i=0}^{\infty} e^{-2i(i+1)t} \frac{\sin\big((2i+1)\theta/2\big)}{\sin(\theta/2)} \quad (5)$$

where $\theta$ is the angle between $x$ and $x_0$. This is also known as Isotropic Gaussian SO(3), $\mathcal{IG}_{SO(3)}$, whose axis-angle representation of rotation naturally aligns with.

However, this formulation introduces several critical limitations that impede the training of protein generation models. First, transforming clean rotations into noisy samples requires either simulating a computationally intensive geodesic random walk using the SO(3) exponential map or performing time-consuming kernel and score precomputations, as implemented in FrameDiff. Furthermore, when $t \to 0$, the heat kernel scaling terms $e^{-\lambda_i t}$ decay slowly despite $\lambda_i \to \infty$, necessitating the evaluation of thousands of terms. This not only increases computational overhead but also faces numerical instability. And therefore, results in undesignable samples if no auxiliary loss is introduced, as shown in Fig. 2.

As such, some protein generation methods resort to a truncated noise schedule to control errors when the noise level is relatively small. However, the diffusion training is thus incomplete and must introduce additional heuristic loss to help the model learn the accurate structure, which introduces biases to the distribution. Other works (Yim et al., 2023a; Bose et al., 2024) turn to Riemannian Flow Matching Models, since they do not require heat-kernel simulations. But more recent work (Lou et al., 2023) demonstrates that RFMs break the vector field's smoothness assumption, which leads to truncated precision when sampling.

## 3 METHOD

We present RoSE, a **Ro**bust **SE**(3)-based Diffusion Model for protein backbone generation via stable score estimation, to achieve stable protein backbone generation. First, we describe the improved score approximation strategy to handle extremely unstable score approximations when t is small (Sec. 3.1). Then, we present our neural network architecture to directly learn the score function using ScoreUpdate trunk (Sec. 3.2). Moreover, we introduce our training objective loss involving score matching and structure auxiliary loss (Sec. 3.3). Lastly, we introduce our sampling procedure. (Sec. 3.4)

### 3.1 ROBUST HEAT KERNEL ESTIMATION ON SO(3)

In this section, we highlight how to control errors when $t$ is small. To efficiently capture small-time diffusion behavior, we leverage **Varadhan's formula** (Varadhan, 1967), which directly relates the heat kernel's asymptotic decay to the manifold's geometry through geodesic distance:

$$\lim_{t \to 0} t \log p_t(x, y) = -\frac{d(x, y)^2}{4}. \quad (6)$$

This establishes a direct link between probability density and geodesic distance, bypassing the infinite sum of eigenfunctions and therefore providing a theoretical foundation for stable approximations. Building upon Varadhan's asymptotic analysis of the heat kernel at small time scales, we adopt an approximation based on geodesic distance. For two points $x, y$ on the SO(3) space, the heat kernel takes the form:

$$p(x_t|x_0, t) = \frac{1}{(2\pi\sigma^2)^{3/2}} \exp\left(-\frac{d(x_t, x_0)^2}{2\sigma^2}\right), \quad (7)$$

where $d(\cdot, \cdot)$ denotes the geodesic distance on the manifold (the rotation angle on SO(3))and $\sigma$ controls the diffusion scale, satisfying $\sigma = \sqrt{2t}$. And $1/(2\pi\sigma^2)^{3/2}$ is the normalizing factor that ensures the density accumulation is 1. This formulation directly stems from the dominant term in Varadhan's formula when $t \to 0$.

However, when $t$ gets larger, Varadhan's formula quickly becomes inaccurate. Therefore, we generalize Varadhan's formula to the **Wrapped Heat Kernel** by explicitly summing over all periodic geodesic replicas:

$$p(x_t|x_0, t) \propto \sum_{k \in \mathbb{Z}^d} \exp\left(-\frac{\|d(x_t, x_0)^2 + 2\pi k\|^2}{2\sigma^2}\right) \quad (8)$$

Truncating the summation to $|n| \leq N$ (typically $N = 3$) achieves a practical balance between computational efficiency and accuracy. When $t$ approaches 0, the wrapped heat kernel degenerates to a main component Gaussian which exactly agrees with Varadhan's formula.

Therefore, we integrate these approximations into a single estimator. Our method partitions the estimation based on the value of $t$, dynamically switching between an eigenfunction summation (for large $t$) and a wrapped heat kernel (for small $t$), and especially Varadhan's formula (for $t \to 0$). This strategy ensures both computational efficiency and controlled approximation errors across various scales $t$. Our algorithm goes as follows:

---

**Algorithm 1:** Integrated Heat Kernel Computation

---

1 **Hyperparameters:** time cutoff $t_c$, truncated parameter $k$
2 **Input:** clean sample $x_0$, time $t$, noised sample $x_t$
3 Compute
4 **if** $t < t_c$ **then**
5      **return** $p(x_t|x_0, t) = 1/C \sum_{k \in \mathbb{Z}^d} \exp\left(-\frac{\|d(x_t, x_0)^2 + 2\pi k\|^2}{2\sigma^2}\right)$, truncated to $|k| < 3$
6 **else**
7      **return** $p(x_t|x_0, t) = \frac{1}{8\pi^2} \sum_{i=0}^{\infty} e^{-2i(i+1)t} \frac{\sin\left((2i+1)d(x_t, x_0)^2/2\right)}{sin(d(x_t, x_0)^2/2)}$, truncated to $|i| < 10$.
8 **end**
9 **remark** $\nabla_x \log p$ can be computed with autodifferentiation.

---

As shown in Fig. 3, our estimation exhibits more stable results than the eigenfunction's computation used in Framediff (Yim et al., 2023c) at small $t$, resulting in a more numerically stable and precise score approximation for the diffusion model.

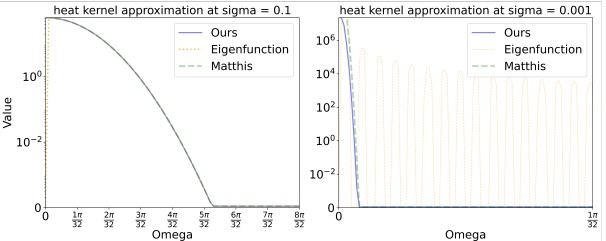

Figure 3: **Heat kernel value comparison between different method.** Blue: Our proposed method. Yellow: Eigenfunction method used in Framediff. Green: Matthis's approximation, used as a reference (see Appendix B in details).

### 3.2 MODEL ARCHITECTURE

Here, we outline our score prediction module, which utilizes cutting-edge SE(3)-equivariant neural networks. The architecture incorporates Invariant Point Attention (IPA) (Jumper et al., 2021) blocks to iteratively refine SE(3) transformations across $L$ layers through a combination of spatial and sequential attention mechanisms. At the $\ell$-th layer, the node features are represented as $h_\ell = [h_\ell^1, \ldots, h_\ell^N] \in \mathbb{R}^{N \times D_h}$, where $h_\ell^n$ denotes the embedding for the $n$-th residue. Simultaneously, the edge features $z_\ell \in \mathbb{R}^{N \times N \times D_z}$ store pairwise interactions, with $z_\ell^{nm}$ encoding the edge between residues $n$ and $m$.

Fig. 4 illustrates a single layer of RoSE, a score-driven geometric graph network that jointly models node relationships and SE(3) scores via dedicated update mechanisms. The input consists of initial node features $h_0$, encoding protein residue indices(sequential positional information), which are first projected linearly and then processed by Invariant Point Attention (IPA) to maintain geometric equivariance. Node features are refined through a NodeUpdate module integrating IPA with Transformer layers, while edge features ($z_l$) are updated via a separate EdgeUpdate. Unlike FrameDiff, which relies on simulation to derive scores from predicted rigid transformations, introducing error propagation, our architecture directly predicts and iteratively optimizes scores through an MLP branch. Additionally, akin to FrameDiff, our model predicts torsion angles for each residue's oxygen atom relative to the predicted local frame.

### 3.3 OBJECTIVE FUNCTIONS

In Eq. 3 and Eq. 4, the reverse process of a Stochastic Differential Equation (SDE), where the score term $\nabla_X \log p_t(X_0|X_t, t)$ is intractable. To address this, we train our score network $s_\theta(X_t, t)$ as a direct score estimator by minimizing the Denoising Score Matching (DSM) loss (Song et al., 2021):

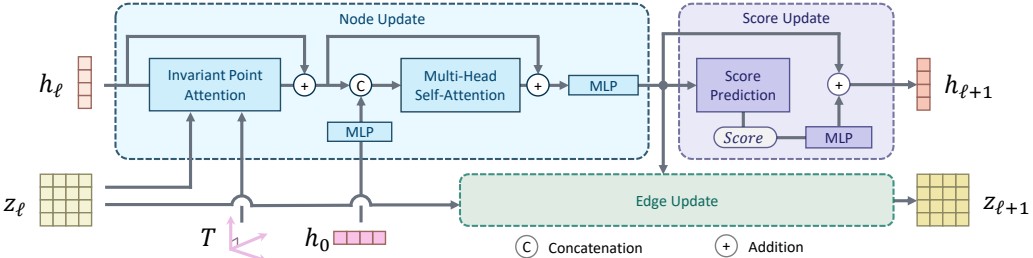

Figure 4: **Single layer of RoSE.** The layer receives the current node features $h_\ell$, the current edge features $z_\ell$, the initial node features $h_0$, and the initial frame representations $T$. Rectangles denote trainable neural networks; italic labels denote layer inputs and outputs. Within the layer, node features are first updated via IPA and multi-head self-attention, producing $h_{\ell+1}$. Updated node features are then used to update the edge features to $z_{\ell+1}$, and in parallel, a small prediction head directly computes the layer's score output.

$$\mathcal{L}(\theta) = \mathbb{E}\left[\lambda_t \|\nabla_X \log p(X_t|X_0, t) - s_\theta(X_t, t)\|^2\right].\tag{9}$$

Given the independence between $\mathrm{SO}(3)$ and $\mathbb{R}^3$, we decompose the DSM loss into separate terms for rotation and translation:

$$\mathcal{L}_{dsm} = \alpha_1 \mathcal{L}_{dsm}^{\mathrm{SO}(3)} + \alpha_2 \mathcal{L}_{dsm}^{\mathbb{R}^3},\tag{10}$$

where $\alpha_1$ and $\alpha_2$ are weighting coefficients for the respective components.

While the DSM loss alone is sufficient for training our protein model, we incorporate an auxiliary loss for ablation analysis. Computing this loss requires reconstructing predicted frames from the estimated scores, which introduces a technical complication: as detailed in Algo. 1, our score computation involves autodifferentiation via torch and disrupts gradient flow during backpropagation. (See Appendix D.)

Here, we alternatively relate score prediction to backbone prediction via Varadhan's formula for small t, given Eq. 6,

$$\nabla_x \log p(x|x_0, t) \approx \frac{1}{2t} \log_x(x_0) \Rightarrow x_0 \approx \exp_x(2t \cdot \nabla_x \log p(x|x_0, t))\tag{11}$$

For small $t$, we incorporate auxiliary losses, including the backbone position loss ($\mathcal{L}_{\mathrm{bb}}$) and the pairwise atomic distance loss ($\mathcal{L}_{\mathrm{2D}}$), in addition to the main diffusion loss ($\mathcal{L}_{\mathrm{dsm}}$), as implemented in FrameDiff. See details in Appendix D.

### 3.4 SAMPLING

We present our sampling procedure on the SE(3) manifold. Using an Euler-Maruyama solver with geodesic random walk, we initialize frames from an invariant density at time $T$: for translations, we sample from a standard Gaussian in $\mathbb{R}^3$, while rotations are drawn from a wrapped Gaussian distribution on SO(3). The reverse-time sampling progresses from $t = T$ to $t = 0$ in discrete steps of size $\eta$. At each step, our model directly predicts the score function $\nabla_X \log p(X_0|X_t, t)$, and we update the rigid frames through exponential mapping of $\{X_t, \mathrm{d}t\}$.

Unlike FrameDiff – which suffers from numerical instability as $t \to 0$ due to score prediction errors and must resort to early truncation ($\epsilon > 0$) and noise downscaling – our method maintains stable generation throughout the complete diffusion process. We achieve this via RoSE score prediction network, trained with our robust score estimator to prevent instability at small $t$.

Table 2: **Quantitative Comparison.** RoSE consistently achieves best or second-best performance in terms of quality, diversity, and novelty. (↓) indicates lower is better, (↑) indicates higher is better.

| Method | Quality | | Diversity | | Novelty | SS Ratio |
|--------|---------|---------|-----------|---------------|---------|----------|
| | scRMSD (↓) | scTM (↑) | inter-TM(↓) | max-cluster(↑) | pdbTM(↓) | ($\alpha/\beta$/coil) |
| RFDiffusion | 1.84 | **0.95** | **0.44** | 0.60 | 0.53 | 82.1/11.6/6.3 |
| FrameDiff | 2.72 | 0.87 | 0.57 | 0.29 | 0.47 | 71.5/24.5/3.9 |
| FrameDiff w/o aux | 12.42 | 0.16 | – | – | – | – |
| FrameFlow | 2.25 | 0.76 | 0.62 | 0.74 | 0.62 | 67.4/26.4/6.1 |
| CarbonNovo | 1.94 | 0.86 | 0.61 | 0.73 | 0.81 | 65.8/20.3/13.9 |
| FoldFlow | 2.87 | 0.63 | 0.74 | 0.41 | 0.65 | 75.2/16.8/8.0 |
| FoldFlow2 | 1.74 | 0.99 | 0.66 | 0.68 | 0.47 | 77.5/14.1/8.4 |
| Protina | 1.79 | 0.98 | 0.62 | 0.76 | 0.52 | 71.4/15.6/12.9 |
| RoSE w/o aux | 1.91 | 0.89 | 0.62 | 0.54 | **0.31** | 67.7/18.2/14.1 |
| RoSE | **1.83** | 0.93 | 0.53 | **0.78** | 0.43 | 63.6/22.9/13.5 |

Table 3: **Motif-scaffolding comparison.**

| Benchmark | RFDiffusion benchmark | | VHH benchmark | | |
|-----------|-----------------------|------------|---------------|-----------|-------------|
| **Model** | **Solved /24 ↑** | **Diversity ↑** | **Motif ↓** | **Scaffold ↓** | **Solved /25 ↑** |
| RFDiffusion | 24 | 0.427 | 3.10 | 2.58 | 5 |
| FrameDiff | 18 | 0.311 | – | – | – |
| FrameFlow | 22 | 0.335 | – | – | – |
| FoldFLow2 | 24 | 0.395 | 2.91 | 1.94 | 7 |
| Ours | 23 | 0.412 | 3.52 | 2.45 | 4 |

## 4 EXPERIMENTS

**Customized Dataset.** Our model was trained on structural data curated from PDB (wwPDB consortium, 2019). We selected monomer proteins with lengths ranging from 60 to 384 residues and applied a radius of gyration filter based on the empirical scaling law observed for globular proteins. Specifically, we retained proteins with a radius of gyration less than $2.24 \times N^{0.392}$Å, where $N$ represents the number of residues in the protein. Additionally, we performed other rigorous quality filtering steps, resulting in a final dataset of 87,426 monomer protein entries. Please see Appendix C for more details.

**Baselines.** We establish comparative baselines across two protein design paradigms: (1) For unconditional generation, we leverage pre-trained implementations of **FrameDiff** (Yim et al., 2023c), **FrameFlow** (Yim et al., 2023a), **CarbonNovo** (Ren et al., 2024), **FoldFlow** (Bose et al., 2024), and **RFDiffusion** (Watson et al., 2022) which serves as the state-of-the-art reference. (2) For motif-scaffolding, we benchmark against conditional implementations of **FrameFlow**, **FrameDiff**, **Protina**(Geffner et al., 2025), **FoldFlow2** (Huguet et al., 2024) and **RFDiffusion**.

### 4.1 UNCONDITIONAL PROTEIN BACKBONE GENERATION

The goal of unconditional protein backbone generation is to produce 3D structures aligning with curated data distribution. We evaluate the generated proteins in terms of *quality*, *novelty* and *diversity*, we kindly refer to Appendix E for detailed definitions

We evaluate two model variants: one without auxiliary losses (**RoSE w/o aux**) and one incorporating Varadhan-based auxiliary losses (**RoSE**). Both are compared against leading SE(3)-based generative models, including RFDiffusion, FrameDiff, FrameFlow, CarbonNovo, FoldFlow, FoldFlow2, and Protina.

As shown in Table 2, our full model achieves competitive or state-of-the-art performance across nearly all metrics. It attains the best scRMSD (1.83↓) and second-best scTM (0.93↑), while also leading in diversity (max-cluster: 0.78↑) and showing strong novelty (pdbTM: 0.43↓). Even without auxiliary losses, our model remains competitive in quality (scRMSD 1.91↓, scTM 0.89↑), confirming the robustness of the core architecture. Notably, RFDiffusion's marginally higher scTM (0.95↑) may be attributed to the higher proportion of alpha-helical structures in its training dataset, as indicated by

Table 4: **Ablation study on score update and heat kernel estimation.** Comparison of RoSE with FrameDiff variants under identical training settings and dataset.

| Method | Quality | | Diversity | | Novelty | SS Ratio |
|---|---|---|---|---|---|---|
| | Designability ($\uparrow$) | scRMSD ($\downarrow$) | scTM ($\uparrow$) | inter-TM ($\downarrow$) | pdbTM ($\downarrow$) | ($\alpha/\beta$/coil) |
| Vanilla FrameDiff | 0.45 | 2.44 | 0.84 | 0.71 | 0.49 | 71.5/24.5/3.9 |
| FrameDiff + Score Update | 0.32 | 3.61 | 0.83 | 0.78 | 0.61 | 66.5/17.1/16.4 |
| RoSE (FrameDiff dataset) | **0.96** | **1.86** | **0.91** | **0.55** | **0.47** | 67.5/18.8/13.7 |

Table 5: **Ablation study on dataset composition.**

| Method | Designability ($\uparrow$) | scRMSD ($\downarrow$) | scTM ($\uparrow$) | inter-TM ($\downarrow$) | Novelty ($\downarrow$) |
|---|---|---|---|---|---|
| Vanilla FrameDiff | 0.48 | 2.44 | 0.84 | 0.61 | 0.49 |
| FrameDiff + Our data | 0.51 | 2.31 | 0.86 | 0.68 | 0.47 |
| RoSE + FrameDiff data | 0.96 | 1.86 | 0.91 | 0.55 | 0.47 |
| RoSE + Our data | **0.98** | **1.81** | **0.96** | **0.51** | **0.44** |

its secondary structure composition (82.1% $\alpha$-helix). Detailed per-length sample results and extended baselines are provided in the Appendix F.

These improvements stem from our stabilized SE(3) score approximation via Varadhan's formula, which ensures training stability and provides geometrically faithful manifold guidance, enabling high-quality, diverse backbone generation.

### 4.2 MOTIF SCAFFOLDING

In protein design, motif scaffolding addresses the challenge of constructing structural frameworks ("scaffolds") around predefined functional segments ("motifs") while preserving their biological activity. This approach allows for the creation of proteins with predetermined functional sites through conditional generation. The motifs, often small and geometrically diverse, require models to incorporate both structural and chemical information for effective scaffolding. Following established evaluation protocols, we employ two distinct benchmarks: (1) the established 24 single-chain motif dataset (Watson et al., 2022), and (2) a newly developed benchmark based on Complementarity Determining Regions (CDRs) in VHH nanobodies, curated from structural antibody databases (Huguet et al., 2024). This evaluation shown in Tab. 3 demonstrates our model's adaptability for conditional generation tasks in protein design.

**Benchmark Results.** Our method demonstrates competitive performance across both benchmarks. In the single-chain motif benchmark, we achieve a 23/24 success rate, matching the state-of-the-art, while maintaining high structural diversity (0.412). On the VHH nanobody benchmark, our approach achieves a 4/25 success rate, which, while lower than FoldFlow2's 7/25, still demonstrates strong scaffolding capabilities.

### 4.3 ABLATION STUDY

#### 4.3.1 SCORE UPDATE AND HEAT KERNEL ESTIMATION

We conducted an ablation study to evaluate the individual contributions of our proposed components. As shown in Table 4, we compare three configurations under identical training settings: (1) **Vanilla FrameDiff**—the original implementation; (2) **FrameDiff + Score Update**—augmented with our score prediction module; and (3) **RoSE**—our complete model incorporating both score update and heat kernel estimation.

The results demonstrate that while directly applying the score update module alone degrades performance (Designability: 0.32 vs. 0.45), the combination with our heat kernel estimation in RoSE significantly improves all metrics, achieving near-perfect designability (0.96) and superior structural quality. This confirms that accurate score estimation via Varadhan's formula is essential for stable training and effective generation.

#### 4.3.2 DATASET EVALUATION

Table 5 evaluates the impact of dataset composition. Our customized dataset provides moderate improvements to FrameDiff (Designability: 0.51 vs. 0.48), but when combined with RoSE, it

Table 6: **Ablation study on sampling steps.**

| Sampling Steps | 50 | 100 | 200 | 300 | 400 | 500 | 600 |
|---|---|---|---|---|---|---|---|
| scRMSD ($\downarrow$) | 18.42 | 7.52 | 3.25 | 2.01 | 1.89 | 1.91 | **1.86** |
| scTM ($\uparrow$) | 0.14 | 0.31 | 0.64 | 0.87 | **0.98** | 0.97 | 0.98 |

achieves the best overall performance across all metrics. This demonstrates that while our architectural innovations drive the majority of performance gains, the curated dataset provides additional refinement. To promote reproducibility, we will release both the dataset and processing scripts.

### 4.3.3 SAMPLING STEP ANALYSIS

We further analyze the effect of sampling steps on generation quality. As shown in Table 6, performance improves steadily with increased sampling steps, with optimal results achieved around 400-600 steps. While our method maintains stable performance across a wide range of step configurations, we set the sampling step to 400 in our experiments to achieve the best trade-off between computational cost and generation quality.

## 5 RELATED WORK

**Protein Generative Models.** Diffusion models have achieved remarkable success in image and video generation (Ho et al., 2020; Song et al., 2021; Ho et al., 2022; Saharia et al., 2022; Zhang et al., 2023; Karras et al., 2024; Brooks et al., 2024), and are increasingly applied to protein structure generation (Watson et al., 2022; Yim et al., 2023c). Early approaches represented proteins via pairwise features and trained Euclidean diffusion models in this feature space (Wu et al., 2022a; Lee et al., 2023), but lacked end-to-end structure generation. Recent methods model protein backbones as rigid frames in SE(3) space (Watson et al., 2022; Yim et al., 2023c;b), incorporating structural priors while facing numerical instability. Alternatively, some apply diffusion directly in $\mathbb{R}^N$ at the atomic level (Trippe et al., 2023; Ingraham et al., 2023; Geffner et al., 2025), but often sacrifice structural priors or restrict modeling scope. Our method preserves SE(3)-based structural modeling while enabling accurate and stable score estimation directly in SE(3) space

**Riemannian Diffusion Models.** With the rise of diffusion models enabling efficient, simulation-free training in Euclidean space (Ho et al., 2020; Song et al., 2021), recent works have extended these approaches to non-Euclidean manifolds. Riemannian diffusion models (RDMs) (Huang et al., 2022) generalize continuous-time diffusion to arbitrary Riemannian manifolds, but face limitations including the lack of closed-form solutions for manifold Ornstein–Uhlenbeck analogs and costly score matching procedures (Chen & Lipman, 2023), hindering scalability to high-dimensional settings such as protein modeling. Scaling Riemannian Diffusion Models (Lou et al., 2023) improve efficiency via a differential-equation-based framework, yet are only validated on low-complexity synthetic data. Building on this line, we extend Riemannian diffusion to SE(3)$^N$, delivering a numerically stable and scalable model capable of learning complex protein manifolds.

## 6 DISCUSSION

**Limitations.** Despite the promising results, our current protein generation pipeline has several limitations. First, we restrict training to high-quality PDB entries to ensure data reliability and stable convergence; in contrast, recent works (Lin et al., 2024; Geffner et al., 2025) have incorporated large-scale resources such as AFDB, which could increase dataset size by over an order of magnitude and potentially improve performance, making integration of such data an important direction for future work. Second, our current evaluation lacks experimental (wet-lab) validation, and the biological viability of generated proteins remains to be confirmed, with bridging this gap between in silico modeling and real-world application being a key avenue for future research.

**Conclusion.** In this work, we introduce a novel method that accurately models the distribution of protein backbones and generates high-quality protein structures. Our enhanced SE(3)diffusion framework enables stable training directly in the protein SE(3) space, optimized via score matching loss without requiring auxiliary terms. We empirically validate the accuracy of our likelihood estimation and demonstrate the model's capability to generate both diverse and designable protein samples. Furthermore, our method exhibits probabilistic scaffolding capabilities, successfully addressing several challenging scaffolding tasks.

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

# A FROM EUCLIDEAN SPACE TO RIEMANNIAN SPACE: A SHORT REVIEW OF SCORE-BASED DIFFUSION MODELS

## A.1 EUCLIDEAN DIFFUSION MODELS

Diffusion models construct generative processes through the interplay of forward noising and reverse denoising dynamics. Consider a data distribution $p_{\text{data}}(\mathbf{y}_0)$ on $\mathbb{R}^D$ that evolves through a continuous-time stochastic process $\{\mathbf{y}_t\}_{t\in[0,T]}$ governed by:

$$d\mathbf{y}_t = \boldsymbol{\alpha}(t)\mathbf{y}_t dt + \boldsymbol{\beta}(t)d\mathbf{W}_t, \quad \mathbf{y}_0 \sim p_{\text{data}} \tag{12}$$

where $\boldsymbol{\alpha}(t) : [0,T] \to \mathbb{R}^{D\times D}$ is a time-dependent drift matrix, $\boldsymbol{\beta}(t) : [0,T] \to \mathbb{R}_+$ is a diffusion coefficient, and $\{\mathbf{W}_t\}$ is $D$-dimensional Brownian motion. The transition density $p_{t|s}(\mathbf{y}_t|\mathbf{y}_s)$ for $0 \leq s < t \leq T$ admits the closed-form solution:

$$p_{t|0}(\mathbf{y}_t|\mathbf{y}_0) = \mathcal{N}(\mathbf{y}_t; \mathbf{A}(t)\mathbf{y}_0, \boldsymbol{\Sigma}(t)) \tag{13}$$

where $\mathbf{A}(t) = \exp\left(\int_0^t \boldsymbol{\alpha}(s)ds\right)$ and $\boldsymbol{\Sigma}(t) = \int_0^t \exp\left(2\int_s^t \boldsymbol{\alpha}(r)dr\right)\boldsymbol{\beta}(s)^2 ds \cdot \mathbf{I}_D$.

The reverse-time process $\{\mathbf{y}_{T-t}\}_{t\in[0,T]}$ follows the stochastic differential equation:

$$d\mathbf{y}_t = \left[\boldsymbol{\alpha}(T-t)\mathbf{y}_t - \boldsymbol{\beta}(T-t)^2\nabla_{\mathbf{y}_t}\log p_{T-t}(\mathbf{y}_t)\right]dt + \boldsymbol{\beta}(T-t)d\widetilde{\mathbf{W}}_t \tag{14}$$

We approximate the score function $\nabla_{\mathbf{y}}\log p_t(\mathbf{y})$ using a neural network $\mathbf{u}_\theta(\mathbf{y},t)$ trained via:

$$\mathcal{L}_{\text{DSM}}(\theta) = \mathbb{E}_{t\sim\mathcal{U}(0,T)}\mathbb{E}_{\mathbf{y}_0}\mathbb{E}_{\mathbf{y}_t|\mathbf{y}_0}\left[\omega(t)\|\mathbf{u}_\theta(\mathbf{y}_t,t) - \nabla_{\mathbf{y}_t}\log p_{t|0}(\mathbf{y}_t|\mathbf{y}_0)\|^2\right] \tag{15}$$

where $\omega(t)$ is a weighting function typically chosen as $\omega(t) = \boldsymbol{\beta}(t)^2$.

## A.2 REMANNIAN DIFFUSION MODELS

Diffusion models on Riemannian manifolds $\mathcal{M}$ extend Euclidean score-based generative modeling to curved spaces. The framework consists of three fundamental components:

[Riemannian Diffusion Process] For a $d$-dimensional compact connected Riemannian manifold $\mathcal{M}$ isometrically embedded in $\mathbb{R}^n$, we have:

1. A forward noising process $(\mathbf{X}_t)_{t\in[0,T]}$ governed by:

$$d\mathbf{X}_t = b(\mathbf{X}_t,t)dt + g(t)d\mathbf{B}_t^{\mathcal{M}} \tag{16}$$

2. A time-reversed denoising process $(\mathbf{Y}_t)_{t\in[0,T]} = (\mathbf{X}_{T-t})_{t\in[0,T]}$ satisfying:

$$d\mathbf{Y}_t = [b(\mathbf{Y}_t, T-t) - g(T-t)^2\nabla\log p_{T-t}(\mathbf{Y}_t)]dt + g(T-t)d\widetilde{\mathbf{B}}_t^{\mathcal{M}} \tag{17}$$

3. A probability flow ODE enabling deterministic sampling:

$$\frac{d\mathbf{Y}_t}{dt} = b(\mathbf{Y}_t, T-t) - \frac{1}{2}g(T-t)^2\nabla\log p_{T-t}(\mathbf{Y}_t) \tag{18}$$

# B EIGENFUNCTIONS AND MATTHIS'S HEAT KERNEL ESTIMATION ON SO(3) GROUP

The heat kernel on SO(3) admits two conventional mathematical representations:

$$f_\epsilon(\omega) = \sum_{\ell=0}^\infty (2\ell+1)\exp(-\ell(\ell+1)\epsilon^2)\frac{\sin((\ell+1/2)\|\omega\|)}{\sin(\|\omega\|/2)} \tag{19}$$

For concentrated distributions ($\epsilon < 1$), the Matthies approximation offers a closed-form solution:

$$f_\epsilon(\omega) \approx \sqrt{\pi}\epsilon^{-3/2}e^{\frac{\pi}{4}-\frac{\|\omega\|^2}{4\epsilon}}\left(\frac{\|\omega\| - e^{-\frac{\pi^2}{\epsilon}}[(\|\omega\| - 2\pi)e^{\pi\|\omega\|/\epsilon} + (\|\omega\| + 2\pi)e^{-\pi\|\omega\|/\epsilon}]}{2\sin(\|\omega\|/2)}\right) \quad (20)$$

**Implementation Note:** The Matthies approximation (Eq. 20), while analytically exact, suffers from numerical instability in finite-precision `tensor` arithmetic. The competing exponential terms $e^{-\pi^2/\epsilon}$ (underflow) and $e^{\pi\|\omega\|/\epsilon}$ (overflow) produce NaN values when $\epsilon < 0.1$ in standard floating-point implementations. Therefore, high-precision ground truth computation using Python's `decimal` module (153-bit precision).

## C  DATASET CONSTRUCTION

Our training data is curated from PDB (wwPDB consortium, 2019). We selected monomer proteins. To obtain high quality structures, we perform following filters:

- structure determination method is X-ray crystallography or 3D electron microscopy,

- resolution less than 4Å,

- radius of gyration ($R_g$) less than $2.24 \times N^{0.392}$Å, where $N$ represents the number of residues in the protein,

resulting in a final dataset of 87,426 monomer protein entries.

## D  OBJECTIVE FUNCTION DETAILS

### D.1  DENOISING SCORE MATCHING LOSS

We train our score network $s_\theta(X_t, t)$ as a direct score estimator by minimizing the Denoising Score Matching (DSM) loss

$$\mathcal{L}_{dsm} = \alpha_1\mathcal{L}_{dsm}^{\mathrm{SO(3)}} + \alpha_2\mathcal{L}_{dsm}^{\mathbb{R}^3}, \quad (21)$$

where

$$\begin{cases} \mathcal{L}_{dsm}^{\mathrm{SO(3)}} = \frac{1}{N}\sum_{n=1}^{N}\|\nabla_{\mathrm{R}}\log p(\mathrm{R}_t^i|\mathrm{R}_0^i, t) - s_\theta(\mathrm{R}_t^i, t)\|^2 \\ \mathcal{L}_{dsm}^{\mathbb{R}^3} = \frac{1}{N}\sum_{n=1}^{N}\|\nabla_{\mathrm{T}}\log p(\mathrm{T}_t^i|\mathrm{T}_0^i, t) - s_\theta(\mathrm{T}_t^i, t)\|^2 \end{cases} \quad (22)$$

where $X = [X^1, \cdots, X^N] \in \mathrm{SE}(3)^N$, $X^i = [\mathrm{R}^i, \mathrm{T}^i]$ represents $i$'th residue's rotation $\mathrm{R} \in \mathrm{SO}(3)$ and translation $\mathrm{T} \in \mathbb{R}^3$, Groud truth rotation score $\nabla_{\mathrm{R}}\log p(\mathrm{R}_t^i|\mathrm{R}_0^i, t)$ are calculated by Algorithm1 and translation score $\nabla_{\mathrm{T}}\log p(\mathrm{T}_t^i|\mathrm{T}_0^i, t) = \frac{\mathrm{T}_t - \sqrt{\bar{\alpha}}\mathrm{T}_0}{\sqrt{1-\bar{\alpha}_t}}$ Song et al. (2021).

### D.2  AUXILIARY FAPE LOSS

While the DSM loss alone is sufficient for training our protein model, we incorporate an auxiliary frame aligned point error (FAPE) loss Jumper et al. (2021) for ablation analysis.

$$\mathcal{L}_{\mathrm{FAPE}} = \frac{1}{N}\sum_{i=1}^{N}\sum_{j=1}^{M}\min\left(d_{\mathrm{cut}}, \left\|\mathrm{X}_i^{-1}(\mathbf{x}_j) - \hat{\mathrm{X}}_i^{-1}(\hat{\mathbf{x}}_j)\right\|_2\right) \quad (23)$$

where $X_i^{-1}$ represents inverse rigid transformation for $i$'th residue. $\mathbf{x}_j$ represents $j$'th atom coordinate.

## E  METRICS

**Quality.** The quality evaluation focuses on the *designability* of protein structures, determined by whether viable amino acid sequences can fold into the generated structures. We employ a computational pipeline where ProteinMPNN (Dauparas et al., 2022) first samples potential amino acid sequences, which are then folded into structures using ESMFold (Lin et al., 2022). The structural similarity between RoSE-generated structures and ESMFold-predicted structures is quantified using TMscore (Zhang & Skolnick, 2004) (sc-TMscore) and RMSD (sc-RMSD), where higher sc-TMscore or lower sc-RMSD values indicate better designability. Additionally, we assess foldability using ESMFold's predicted local distance difference test (pLDDT) score, considering structures with pLDDT > 70 as physically plausible.

**Novelty.** Structural novelty is evaluated by comparing each generated protein against all known structures in the Protein Data Bank (PDB) (wwPDB consortium, 2019). For each generated structure, we compute the TMscore against every PDB entry and record the highest value (pdb-TM). A lower pdb-TM score indicates that the generated structure is more distinct from known natural proteins, representing greater novelty.

**Diversity.** We quantify the diversity of generated protein structures through two complementary approaches. First, we calculate pairwise TMscore similarities between all generated structures, where the maximum pairwise similarity (inner-TM) serves as a diversity metric - lower inner-TM values indicate more diverse structure sets. Second, we perform clustering analysis using FoldSeek (Van Kempen et al., 2024) to determine the number of distinct structural clusters, with diversity measured as the ratio of unique clusters to total structures generated.

**SS Ratio.** The secondary structure (SS) ratio quantifies the proportion of residues in the generated protein structures that adopt regular secondary structure elements, specifically alpha-helices and beta-sheets. We employ DSSP (Kabsch & Sander, 1983) to assign secondary structure annotations and calculate the ratio of residues classified as helical or strand conformations to the total number of residues.

## F UNCONDITIONAL GENERATION

**Comparison on sequence length range.** Our evaluation range for sequence length is consistent with state-of-the-art methods such as FoldFlow (Bose et al., 2024), GAFL (Wagner et al., 2024), and ReQFlow (Yue et al., 2025), all of which, including RoSE, evaluate protein sequences within the length range of 70 to 350 in their main results. However, as requested, we have conducted additional experiments to evaluate baseline performance on longer sequences. We report scRMSD across varied protein lengths as follows:

Table 7: scRMSD across varied protein lengths

| Method | 100 | 200 | 300 | 400 | 500 | 600 |
|---|---|---|---|---|---|---|
| FrameDiff | 1.13 | 1.97 | 2.56 | 3.62 | 8.89 | 11.43 |
| RFDiffusion | 0.99 | 2.11 | 2.31 | 3.41 | 8.14 | 9.41 |
| Frameflow | 1.17 | 2.10 | 2.43 | 4.23 | 9.88 | 12.47 |
| Ours | 1.14 | 1.84 | 1.95 | 2.41 | 4.36 | 7.14 |

**Comparison on model inference.** As shown in Table 8, we present a comparison of model parameters and inference time among different methods.

Table 8: Comparison on model inference

| Method | Model parameters | Inference steps | Inference time (s) | | | | |
|---|---|---|---|---|---|---|---|
| | | | 100AA | 150AA | 200AA | 250AA | 300AA |
| FrameDiff | 17M | 500 | 8 | 10 | 11 | 12 | 14 |
| FrameFlow | 150M | 200 | 3 | 5 | 7 | 7 | 9 |
| FoldFlow2 | 20M | 200 | 5 | 6 | 8 | 9 | 10 |
| RoSE | 23M | 400 | 8 | 9 | 12 | 13 | 14 |

