# OpenReview forum: "RoSE: Enhancing SE(3)-based Protein Backbone Generation via Robust Score Estimation"
_ICLR.cc/2026/Conference — ICLR 2026 Conference Withdrawn Submission_

### Official Review · Reviewer_JJgT · 2025-10-20

**Soundness:** 2
**Presentation:** 2
**Contribution:** 2
**Rating:** 2
**Confidence:** 3

**Summary:**

The paper identifies a key bottleneck in applying Riemannian diffusion models to protein structure generation: as $t \to 0$, score estimation on the manifold $SO(3)$ becomes numerically unstable. To address this, the authors introduce a new heat-kernel approximation that stabilizes and improves the accuracy of $SO(3)$-based diffusion in the small-time regime. The model shows excellent performances compared to some baselines in protein generation.

**Strengths:**

1. It is true that conventional $SO(3)$ diffusion models suffer from numerical instabilities. The authors’ proposed approach is therefore well motivated.
2. The results in experiments is good compared to baselines.

**Weaknesses:**

1. Equation (5) and the equation in Algorithm 1 lost a (2i+1).
2. Equation (7) is theoretically incorrect: on a Riemannian manifold the stated identity does not hold exactly. It is, at best, a small-time ($t \to 0$) approximation, so writing it with an equality sign is not rigorous.
3. Equation (8) and Algorithm 1 contain clear mistakes. Expressions of the form $d(\cdot, \cdot)^2 + 2\pi k$ are ill-defined. $d^2$ and $2\pi k$ are inconsistent metrics. (Do authors mean $d(\cdot, \cdot) + 2\pi k$?). Also, what does \| d^2 \|^2 mean?
4. I also question the fairness of the baseline comparisons. As noted in Appendix F, the models compared to FrameDiff differ in parameter count. Can the authors provide a fairer test, which keep the FrameDiff framework fixed and replace only the $SO(3)$ part with the proposed Rose.
5. I notice that in the Algorithm 1 Line 7 authors write $I<10$. Is it correct? I think in Framediff, $i$ is about 1000.
6. How does the model choose the suitable $t_c$? Can the authors provide some ablation? If $t_c$ is large, (not $t_c \to 0$), the proposed method is incorrect.
7. There are some mistakes in Table 2. e.g. FoldFlow shows the best result in scRMSD and scTM.
8. I am also not clear what is the function of score update module?

**Questions:**

The same as Weaknesses. Many theoretical inconsistencies exist in the proposed method.

---

### Official Review · Reviewer_e5w3 · 2025-10-31

**Soundness:** 3
**Presentation:** 3
**Contribution:** 2
**Rating:** 6
**Confidence:** 4

**Summary:**

This work introduces a novel method that accurately models the distribution of
protein backbones and generates high-quality protein structures. The enhanced SE(3)diffusion framework enables stable training directly in the protein SE(3) space, optimized via score matching loss without requiring auxiliary terms.

**Strengths:**

1. This paper proposes RoSE, a Robust SE(3) diffusion model that leverages Varadhan’s asymptotic formula for the heat kernel tailored for protein design task.
2. This papers successfully applies RoSE, together with architecture improvements and dataset curation to protein structure generation.

**Weaknesses:**

1. The sampling steps required by RoSE is 4 times more than flow-matching-based methods (e.g., frameflow), and the performance drops dramatically in the setting of 50-200, where flow-based methods could handle this fewer steps for efficiency.
2.  Limited evaluation beyond protein structure generation tasks for improvements to Riemannian diffusion models. Including more tasks in original Riemannian diffusion papers will be appreciated. Generally I like the technical and theoretical innovations in AI4Science field.
3. Auxiliary losses are still needed for computational stability. A more careful and quantitive ablation study is appreciated for auxiliary losses.

**Questions:**

1. How to implement the conditional protein generation methods for motif-scaffolding and nanobody design?
2. Could the reliance on auxiliary losses stem from the x0-prediction objective used for framediff? Since it’s basically the structure prediction loss.
3. Will the x0-prediction objective, or more complicated protein structure prediction loss resolve the score prediction instability issues as in framework like flow matching? As in the end, structure prediction is the core ability for AI to generate proteins.

---

### Official Review · Reviewer_mCE1 · 2025-10-31

**Soundness:** 2
**Presentation:** 3
**Contribution:** 2
**Rating:** 2
**Confidence:** 4

**Summary:**

1. The paper addresses a numerical instability issue when $t\to0$ in SE(3)-based diffusion models. The proposed heat kernel estimator combining Varadhan's formula with wrapped heat kernel summation provides a theoretically grounded solution that should be applicable to recent diffusion-based protein generation models. The integration of wrapped heat kernel approximation (Eq. 8) for handling periodic geodesic replicas on SO(3) manifold is a meaningful technical contribution that balances computational efficiency and accuracy.
2. Unlike FrameDiff which derives scores from predicted rigid transformations (introducing error propagation), RoSE directly predicts scores through an MLP branch, which is architecturally cleaner.
3. The paper provides clear mathematical formulation of the problem and the proposed solution is well-motivated with good visual illustrations.

**Strengths:**

1. The paper addresses a numerical instability issue when $t\to0$ in SE(3)-based diffusion models. The proposed heat kernel estimator combining Varadhan's formula with wrapped heat kernel summation provides a theoretically grounded solution that should be applicable to recent diffusion-based protein generation models. The integration of wrapped heat kernel approximation (Eq. 8) for handling periodic geodesic replicas on SO(3) manifold is a meaningful technical contribution that balances computational efficiency and accuracy.
2. Unlike FrameDiff which derives scores from predicted rigid transformations (introducing error propagation), RoSE directly predicts scores through an MLP branch, which is architecturally cleaner.
3. The paper provides clear mathematical formulation of the problem and the proposed solution is well-motivated with good visual illustrations.

**Weaknesses:**

1. Table 2 shows that while the authors frame the comparison as unconditional generation, RoSE actually underperforms both Protina (scRMSD: 1.83 vs. 1.79; scTM: 0.93 vs. 0.98) and FoldFlow2 (1.76, 0.96) on monomer generation—a task both are designed to handle.
2. The authors acknowledge in the Discussion that they restrict training to high-quality PDB entries, while competitive methods like Protina incorporate AFDB, increasing dataset size by over an order of magnitude. This is a critical limitation: the underperformance in Table 2 is likely directly attributable to this restricted training data, yet the authors provide no empirical evidence of scaling to larger datasets. Without demonstrating scalability to large-scale datasets, the practical advantage over existing methods remains unclear.
3. Table 3, the paper emphasizes the VHH nanobody benchmark but achieves only 4/25 success rate compared to FoldFlow2's 7/25. More importantly, for antibody-specific tasks, comparison with specialized antibody models (e.g., IgGM and other CDR-specific design methods) would be more meaningful than comparison with general protein design methods like RFDiffusion. The significance of this benchmark for demonstrating the method's advantages is questionable.
4. While the wrapped heat kernel is a solid technical contribution, the core idea of using Varadhan's formula for small-t approximation has been explored in the Riemannian diffusion literature (Lou et al., 2023). The adaptation to protein design, though practical, represents incremental rather than fundamental innovation.

**Questions:**

1. Can you provide empirical evidence (even preliminary experiments) that RoSE successfully trains on AFDB-scale datasets (500K+ structures)? Does the performance gap with Protina in Table 2 close when both are trained on comparable data? This is crucial for evaluating whether the underperformance is a data issue or a fundamental limitation of the method.
2. Table 4 shows a significant increase in loop/coil proportion (13% for RoSE vs. 3.9% for FrameDiff). This dramatic shift in secondary structure distribution is concerning and warrants deeper investigation. Why does stabilizing the score estimation lead to such a substantial change in structural preferences? Is this a desirable property or an artifact of the training procedure?
3. While mentioned in the metrics section (Appendix E), actual pLDDT values are absent from the main results, making it difficult to comprehensively assess the quality. I suggest the authors also report the designability in Table 2 as they did in Table 4 and Table 5.
4. Despite claiming the model works "without auxiliary terms", Table 2 shows the full RoSE with auxiliary loss performs significantly better than "RoSE w/o aux" (scTM 0.93 vs. 0.89, pdbTM 0.43 v.s. 0.31). Does this indicates auxiliary losses are still important in practice?
5. Line 93-94: The term "timestamp" should be "timestep" throughout the manuscript. This terminology error appears multiple times and should be corrected for clarity.
6. Table 2 Protina and FoldFlow2 outperform RoSE but are not bolded.
7. What is the inference time for a 300aa protein?

---

### Official Review · Reviewer_eUSb · 2025-10-31

**Soundness:** 2
**Presentation:** 2
**Contribution:** 2
**Rating:** 2
**Confidence:** 3

**Summary:**

In this paper, the authors introduce RoSE (Robust SE(3) diffusion model), a diffusion model methodology using Varadhan's formula for the approximation of the heat kernel for SO(3) score estimation. They propose architectural modifications to FrameDiff which improve the model performance. The new method is evaluated on several protein generation benchmarks.

**Strengths:**

* The authors improve the architecture of FrameDiff.

* Experimentation is extensive and seem to produce results on part with the state of the art regarding many different metrics. Ablation studies are extensive.

* I do appreciate the honest "Limitation" paragraph in the conclusion of the paper.

**Weaknesses:**

* The use of Varadhan’s formula in the case of Riemannian diffusion models to stabilize the computation of the score was first presented in [1]. Either the authors are not aware of this work or they deliberately avoided mentioning this existing method. In both cases, I am pushing back on the novelty claims made in that paper. In particular "In this work, we introduce a novel method that accurately models the distribution of protein backbones and generates high-quality protein structures" seems like an inflated claim. I think this misrepresents the work. In my opinion the main improvements regard the architecture and the further stabilization of Varadhan's formula with wrapping.

* The related work section is quite poor and misrepresents the field of Riemannian generative modeling. [1] (which as mentioned contains Varadhan's formula) is not cited nor mentioned in the related work. Flow matching methods which are extremely competitive for those problems is also omitted [2]. The contributions of [3] which also consider a very similar expressions related to a "heat kernel" (see their Equation (3)) are also not considered. Wrapped kernels were also considered in [4] for instance, see Equation (3) (this paper is not cited in the related work as well). It is the duty of the authors to  accurately represent the state of the related work, especially when the paper contains claims of introducing novel methods.

* The math is wrong. The expression of the heat kernel given in (7) is **not** the expression of the heat kernel on a Riemannian manifold. This is a common misconception, the approximation is only valid in the small scale limit (i.e., the Varadhan formula). This is a gross error. We refer the authors to [5] for an introduction to the heat kernel on manifolds. As a result while the formula the authors might be a good approximation of the true heat kernel (I found Figure 3 to be interesting), there is not enough justification for the introduction of this formula and there remains work to be done to provide better explanation for this.

* Experimental are misleading. The claim "As shown in Table 2, our full model achieves competitive or state-of-the-art performance across nearly all metrics. It attains the best scRMSD (1.83↓) and second-best scTM (0.93↑)" is wrong just looking at Table 2. In terms of scRMSD it is behind FoldFlow and LaProtina. Similarly for scTM. This discrepancy is never commented

Overall, while I think the practical contribution is worthwhile the inflated claims, poor related work, mistakes in the theory and incorrect experimental metrics, murky the real contributions of the paper.

[1] De Bortoli et al. (2022) -- Riemannian Score-Based Generative Modelling

[2] Chen et al. (2023) -- Flow Matching on General Geometries

[3] Urain et al. (2023) -- SE(3)-DiffusionFields: Learning smooth cost functions for joint grasp and motion optimization through diffusion

[4] Jing et al. (2022) -- Torsional Diffusion for Molecular Conformer Generation

[5] Saloff-Coste (2010) -- The heat kernel and its estimates

**Questions:**

* How is the critical time computed?

* "Protina" I think the authors refer instead to Proteina [1]. I do not think that this model is predicting the frame. Can the authors then comment on "Both are compared against leading SE(3)-based generative models, including RFDiffusion, FrameDiff, FrameFlow, CarbonNovo, FoldFlow, FoldFlow2, and Protina".

[1] Geffner -- Proteina: Scaling Flow-based Protein Structure Generative Models

---

### Note · Authors · 2025-11-24

**Comment:**

Dear Area Chair and Reviewers,

We have decided to withdraw our submission from ICLR 2025.

We sincerely thank the reviewers for their time and detailed feedback. We are particularly grateful to the reviewers for pointing out the theoretical inaccuracies regarding the heat kernel formulation and the mathematical errors in our equations. We also appreciate the feedback concerning the related work and baseline comparisons.

In our next steps, we plan to rigorously revise the mathematical framework to address the mentioned inconsistencies and ensure theoretical correctness. We will also update the literature review and refine our experimental comparisons based on your constructive suggestions.

Thank you again for your valuable input.

**Withdrawal Confirmation:**

I have read and agree with the venue's withdrawal policy on behalf of myself and my co-authors.